# Sickle cell disease and pregnancy profile of complicated malaria in 982 pregnancies in Kinshasa

Tite Minga Mikobi[1,2,3]*, Nelly Ciombo Kamuanya[1,3], Pierre Zalagile Akilimali[4], Prosper Tshilobo Lukusa[1,2,5]

1 Centre d'Excellence de la Drépanocytose (CED), Molecular Biology Service, Department of Basic Sciences, Faculty of Medicine, University of Kinshasa, Kinshasa, DRC, 2 Center for Human Genetics, Faculty of Medicine, University of Kinshasa, Kinshasa, DRC, 3 Obstetrics Gynecology Department, Sickle Cell Center, Institut de Recherche en Sciences de la Santé, Kinshasa, DRC, 4 Department of Biostatistics, School of Public Health, Faculty of Medicine, University of Kinshasa, Kinshasa, DRC, 5 Genetics Unit, Neonatology Service, Department of Pediatrics, University Hospital of Kinshasa, Faculty of Medicine, University of Kinshasa, Kinshasa, DRC

* tite.mikobi@unikin.ac.cd

**Data Availability Statement:** All relevant data are within the paper and its Supporting information files.

## Abstract

### Introduction

Malaria is associated with high morbidity during pregnancy. Homozygous sickle cell pregnant women are even more exposed during complicated malaria. The objective of the study was to evaluate the maternal and fetal morbidity of homozygous sickle cell pregnant pregnant women with complicated malaria.

### Methods

We conducted a retrospective case-control study of 982 pregnancies in sickle cell pregnant women, during which a group of sickle cell pregnant women who received antimalarial chemoprophylaxis was compared to another group without chemoprophylaxis. We analyzed the clinical evolution of pregnant women (VOCs and transfusions, pregnancy weight gain) and parasite (parasite density at the time of diagnosis of complicated malaria and during treatment for three days). We analyzed the parameters of newborns at birth (age of pregnancy at the time of delivery, birth weight, weight of the placenta and histopathological examination of the placenta.

### Results

Out of 982 pregnancies, 15% of pregnant women suffered from complicated malaria, 57% suffered from uncomplicated malaria and 28% did not suffer from malaria. Pregnancy weight gain, birth weight, was better in the group of pregnant women who received chemoprophylaxis and the placenta had less histological lesions. Parasite density was low. There was a significant positive correlation between parasite density and the number of CVOs and transfusions and between parasite density and histological lesions of the placenta and low birth weight.

**Funding:** The authors received no specific funding for this work.

**Competing interests:** The authors have declared that no competing interests exist.

## Conclusion

Complicated malaria is associated with high maternal and fetal morbidity in sickle cell patients. Malaria chemoprophylaxis can reduce maternal and fetal complications and parasite density during malaria infection.

## Introduction

Malaria is the most widespread parasitic disease in the world in general and in the African intertropical zone in particular [1]. In Central Africa, malaria is a public health problem [2, 3].

Indeed, according to the WHO, the number of malaria cases in 2020 was estimated at 241 million, an increase of 14 million new cases compared to 2019 [4, 5]. During the year 2020, the number of deaths due to malaria was estimated at 62,700, an increase of 6,900 deaths compared to the previous year. Half of the deaths due to malaria during the year 2020 were observed in six countries including the DRC [4, 5]. Children under 5 years and pregnant women are the most vulnerable. In 2020, the mortality of children under five years was 272,000, 94% of which in Africa. Eleven percent of this mortality of children under five years old was observed in the DRC. Malaria is transmitted by the bite of an infected mosquito of the Anopheles genus and more rarely during a blood transfusion or by mother-to-child transmission during pregnancy [6, 7]. *Plasmodium falciparum* is responsible for complicated malaria [8]. It is responsible for half of malaria-related deaths worldwide. The other species including *p. vivax*, *p. oval*, *p. malariae* and *p. knowlesi* are not associated with complicated forms of malaria. Since October 2021, WHO recommends the use of the RTS, S/AS01 vaccine against malaria in all areas with a high endemicity of *P. falciparum* malaria. Malaria control strategies include vector control [9, 10]. This is why the use of impregnated mosquito nets is now recommended by the WHO in endemic areas with a high prevalence of malaria. In the DRC, impregnated mosquito nets are distributed free of charge to pregnant women during prenatal consultations. Antimalarial prophylaxis during pregnancy aims to reduce the risk of complicated malaria during pregnancy [11]. Indeed, malaria is recognized in the tropics as being one of the causes of early abortions, prematurity, fetal death in utero and low birth weight [6, 7, 11–14]. Sickle cell disease (SCD) is a constitutional hemoglobinopathy, with autosomal recessive transmission. It is characterized by the E6V mutation of the beta globin gene, in which a purine base [A] is replaced by a pyrimidine base [T]: A>T. This transversion of the purine base by a pyrimidine base results in the substitution of glutamic acid by valine at position 6 of the beta globin chain [15, 16]. The substitution of a hydrophilic amino acid (glutamic acid) by a hydrophobic amino acid (valine), leads to the synthesis of an abnormal hemoglobin called HbS [17]. Indeed, in concentrated solution and under the influence of a drop in oxygen pressure, HbS undergoes a supramolecular process of polymerization [16, 18]. The HbS mutation is today the most common structural anomaly of hemoglobin in the world [19]. Equatorial Africa is the area with the highest prevalence of SCD and the most severe form in the world. Geographically, the distribution of HbS overlaps with that of malaria [10, 20]. Genetically, there are homozygous subjects (SS) and heterozygous subjects (AS). Homozygotes are the sick, while heterozygotes are carriers of the sickle cell trait. Heterozygous is thought to protect against complicated forms of malaria [10, 20–22]. However, homozygotes are very exposed to complicated forms of malaria [10, 20, 21]. Clinically SCD is characterized by recurrent vaso-occlusive ischemic crisis (VOC), chronic hemolysis and a high susceptibility to infections [23]. When the partial pressure of oxygen drops (hypoxia) and when associated with other

unfavorable conditions such as increased temperature, acidosis, HbS crystallizes and causes deformation of the red blood cell [23]. In Africa, the DRC is the second country most affected by SCD and malaria after Nigeria.

The objective of the present study is to evaluate the fetal maternal mobility of complicated malaria in sickle cell pregnant women.

## Methods

This is a retrospective case-control study that was conducted at sickle cell center of Kinshasa (CMMASS) between 2011 and 2020. The study included 982 sickle cell pregnant women. The sickle cell center of Kinshasa (CMMASS) is a center specializing in the care of sickle cell patients. It is a public center and it has the particularity of receiving many pregnant women with SCD.

### Inclusion criteria

In the present study we included known sickle cell pregnant women with a confirmatory molecular diagnosis of SCD. You must have had complicated malaria during pregnancy. The diagnosis of pregnancy had to be made and confirmed by ultrasound in the first trimester of pregnancy between the 7th and 10th week of amenorrhea. The pregnant woman had to have regularly followed the prenatal consultations at the sickle cell center and had given birth in this center.

### Exclusion criteria

We excluded any pregnant woman who developed pre-eclampsia during pregnancy.

**Study groups.** For 15 years, we have observed that many sickle cell pregnant women who receive free impregnated mosquito nets during prenatal consultations, do not accept preventive antimalarial chemoprophylaxis during pregnancy. Thus in the present study, we have two groups of pregnant women with SCD who presented the complicated malaria during pregnancy. The first group (G1) consisted of sickle cell pregnant women who had accepted preventive antimalarial chemoprophylaxis during pregnancy. The second group (G2) consisted of sickle cell pregnant women who did not accept antimalarial chemoprophylaxis during pregnancy.

### Operational definitions

Uncomplicated malaria: in our study we defined uncomplicated malaria as any case of malaria with a parasite density $< 100$ trophozoites per microliter of blood, fever $< 38°C$ and without any threat to vital functions.

Complicated malaria has been defined on the basis of criteria defined by the WHO, among which there is hyperparasitaemia [24, 25].

Weight gain during pregnancy ($\Delta P$): was calculated by the difference between the weight of the pregnant woman at the time of delivery ($P_f$) and the weight at the start of pregnancy ($P_i$): ($\Delta P = P_f - P_i$).

Low birth weight (LBW): we considered the low birth weight any newborn born at term (37 weeks) with a weight below the 10th percentile (10% of birth weight) i.e. overall a weight below 2500 g.

Preventive chemoprophylaxis is the use of drugs or drug combinations to prevent malaria infection and its consequences. It includes chemoprophylaxis, intermittent preventive treatment for pregnant women.

**Preventive chemo prophylaxis.** In our study we used the association Arthemeter and Lumefantrine as a molecule for preventive chemoprophylaxis. Arthemeter was dosed at 80 mg and Lumefantrine at 480 mg.

## Management pregnant women during pregnancy

During the pregnancy, we had given two doses of chemoprophylaxis to group 1 at the 14th week and at the 28th week. All pregnant women in both groups received regular supplementation with omega 3, magnesium pidolat and folic acid. Any case of complicated malaria during pregnancy was treated with the artesunat administered parenterally. The dose administered was 2.4 mg/Kg on arrival, then successively at the 12th hour, 24th hour, 48th hour and finally at the 72nd hour. After 72 hours, the treatment was relayed orally with the association Arthemeter dosed at 80 mg and lumefantrine dosed at 480 mg for 3 days.

## Laboratory analyzes

**Hemoglobin assay and thick smear analysis.** The determination of hemoglobin and the analysis of the thick film were carried out at each appointment of the prenatal consultation.

## The diagnosis of malaria

The diagnosis of malaria was made on the basis of two tests. The rapid test (RDT) using the immunochromatography technique and the microscopic diagnostic test for malaria. For the microscopic diagnosis of malaria, we had carried out the thin smear and the thick drop. The thin smear was intended to make the diagnosis of the plasmodium species, and the thick smear allowed us to determine the density of the parasites per microliter of blood according to WHO recommendations [24, 25].

## Histopathological analysis of the placenta

The placentas of newborns of mothers who had complicated malaria during pregnancy were analyzed in the anatomopathological department of the University Hospital of Kinshasa. Indeed, after delivery, the placenta was analyzed macroscopically in search of any morphological abnormality.

The umbilical cord was cut at the base. Then we weighed the placenta after cutting the membranes of the placenta. We had taken three samples of the placenta from three different sites: at the base of the cord, from the middle zone of the placenta and from the marginal zone. The biopsy specimens were fixed in 10% buffered formalin and then sent to the pathological anatomy laboratory of the University Hospital of Kinshasa. According to the types of lesions observed, we have classified the placentas into five types as defined below.

Placenta type 1: represents placentas that had no lesions after histopathological analysis.

Placenta type 2: these are placentas that had multiple areas of calcification, infarction and fibrin deposition.

Placenta type 3: these are placentas had lesions of thrombosis of chorio allantoic vessels and chorioangiosis.

Type 4 placenta: these are placentas that had type 3 lesions with ischemic necrosis lesions with deposition of fibrinoid substances.

Placenta type 5: these are placentas that had type 4 lesions with chronic villitis and the presence of parasites: trophozoites.

## Variables of interest

In the present study, the variables of interest were: the age of the pregnant woman, the parity, the hemoglobin level, the gravidic weight gain, the weight of the newborn, the age of pregnancy at delivery, placental weight and histopathological lesions of the placenta.

## Ethical approval

The study was approved by the ethical committee of the school of public health of the University of Kinshasa (Approval reference: ESP/CE/079/2016), DRC. Informed consent was obtained from all patients before their inclusion in the study. The informed consent obtained was written and our study did not include minors.

## Statistical analyzes

Our data was processed with SPSS software. We calculated the frequencies and the means and standard deviations. Our averages were compared using the Student t test. The value of $p < 0.05$ was considered as the level of significance.

## Results

Our study included 982 mothers with SCD, among whom 556 (57%) had presented uncomplicated malaria, 146 (15%) had presented complicated malaria and 280 (28%) had not presented malaria during pregnancy. Fig 1 below shows the distribution of malaria in our study population.

   Table 1 above presents the characteristics of our pregnant women. We observe that there was no difference in age and parity between the mothers of the two groups. However, the difference between the mean Hb level during pregnancy and the gravidic weight gain between the groups was highly significant ($p < 0.001$) in favor of group 1.

   Fig 2 above shows the evolution of the parasite density during the treatment for 3 days. It appears that from the first day to the third the parasite densities of the pregnant women of

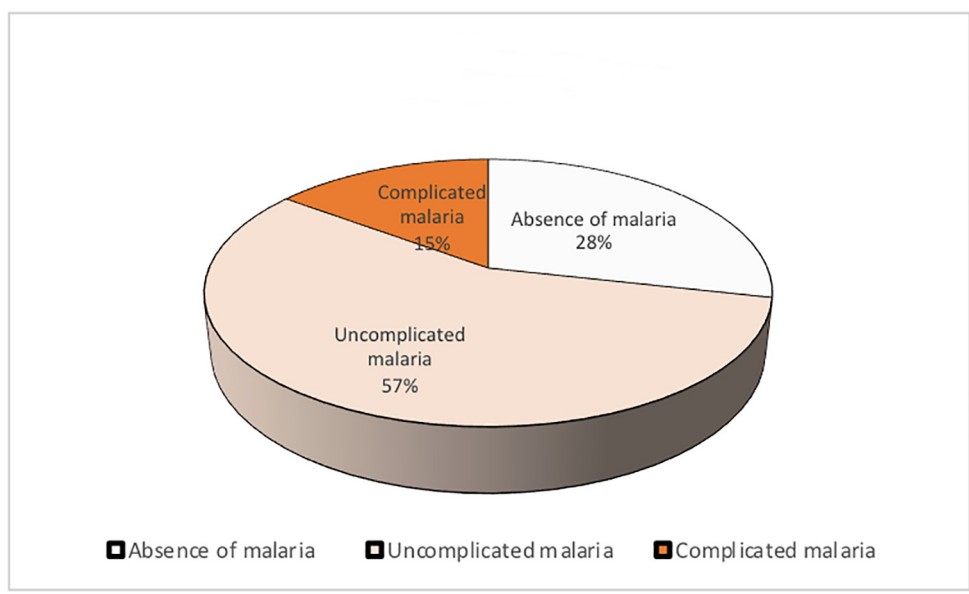

**Fig 1. Distribution of malaria during the pregnancy in study population.**

**Table 1. Periods and therapeutic doses of artemether during complicated malaria.**

| Characteristics | Group 1 (n = 62) | Group 2 (n = 85) | p |
|---|---|---|---|
| Age (years) | 23.60 ± 4.00 | 24.03 ± 3.75 | 0.498 |
| Parity (n) | 1.40 ± 0.52 | 1.30 ± 0.46 | 0.199 |
| Hb (g/dl) | 7.43 ± 1.77 | 5.95 ± 0.98 | <0.001 |
| Gravidic weight gain (Kg) | 5.48 ± 1.11 | 3.19 ± 0.52 | <0.001 |

Group 1: is the study group i.e. pregnant women who had agreed to receive chemoprophylaxis during pregnancy.

Group 2: is the group of cases i.e. pregnant women who did not accept chemoprophylaxis during pregnancy.

group 2 were very high compared to the pregnant women of group 1. The statistical differences were significant (p<0.001).

Table 2 above shows the characteristics of the newborns and the average weight of the placentas. We observe that the mean age at childbirth was similar in the two groups. However, fetal weight, placental weight and frequency of low birth weight were different between the two groups and the differences were statistically significant (p<0.001).

Table 3 shows the frequencies of the types of histopathological lesions observed in the placentas. It appears that the frequency of type 1 placenta (placenta without histological lesion) was higher in group 1 than in group 2. The statistical difference was significant (p<0.001).

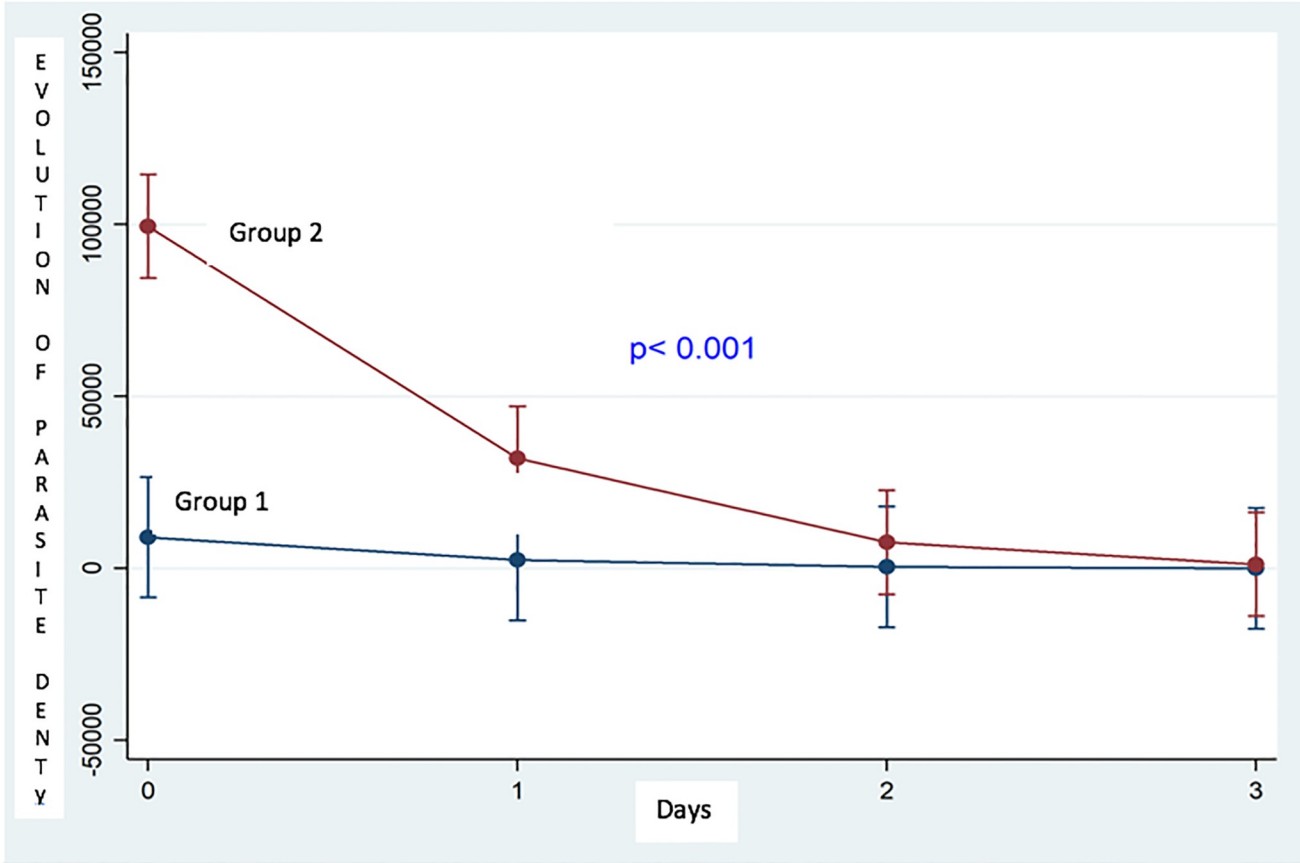

**Fig 2. Evolution of parasite density during treatment in the study population.**

**Table 2. Characteristics of newborns and placentas.**

| characteristics of newborns and placentas | Group 1 (n = 62) | Group 2 (n = 85) | p |
|---|---|---|---|
| Age at childbirth (weeks) | 37.41 ± 0.50 | 37.56 ± 0.52 | 0.042 |
| Fetal weight (gr) | 2721.55± 202.51 | 2503.69 ± 175.66 | <0.001 |
| Placenta weight (gr) | 612.09 ± 148.02 | 438.33 ± 145.27 | <0.001 |
| LBW (%) | 12.90±1.09 | 44.05±1.99 | <0.001 |

LBW: low birth weight

However, the frequency of placenta type 5 (placenta with ischemic necrosis lesions with deposition of fibrinoid substances, with chronic villitis and presence of trophozoites) was higher in group 2, the statistical difference was also significant (p<0.001).

Fig 3 shows the correlation between parasite density and the risk of histological damage to the placenta in our cohort. It appears that there is a positive correlation between the high parasite density and the type 5 placenta (placenta with ischemic necrosis lesions with deposition of fibrinoid substances, with chronic villitis and presence of trophozoites). Conversely, we observe a positive correlation between the low parasite densities with the type 1 placenta (without histological lesion).

Fig 4 shows the frequency of VOCs and transfusions during the malaria attack in the two groups. It appears that during the crisis of complicated malaria, the number of VOCs and transfusions were higher in group 2 than in group 1. The statistical difference was highly significant (p<0.001).

## Discussion

Malaria is a parasitic disease that is associated with great morbidity during pregnancy and in children under five years of age [9, 10, 26–28]. In homozygous sickle cell pregnant women, malaria is responsible for complicated VOCs [29–31] and frequent haemolysis [29–32] which compromises the vital prognosis of the patients [30–32]. During pregnancy, complicated malaria is associated with a high risk of LBW [29–33]. The objective of antimalarial chemoprophylaxis is to prevent malaria infection and its consequences. Our study aimed to evaluate the maternal and fetal morbidity of complicated malaria in sickle cell pregnant women. Our study showed that complicated malaria is frequent during pregnancy. Indeed, 15% of pregnant women in our series suffered from complicated malaria during pregnancy. We have observed that chemoprophylaxis reduces parasite density in complicated malaria. This observation proves the protection of antimalarial chemoprophylaxis during pregnancy. In general, pregnant women with SCD have low weight gain during pregnancy and their newborns have LBW [29–33]. Our study showed that complicated malaria increases the risk of LBW. Indeed, in our series, pregnancy weight gain and birth weight were lower in pregnant women who had not

**Table 3. Frequencies of placental lesions.**

| type of histopathological lesions of the placenta | Group 1 (n = 62) | Group 2 (n = 85) | p |
|---|---|---|---|
| Type 1 (%) | 59.68 | 23.81 | <0.001 |
| Type 2 (%) | 24.19 | 19.05 | 0.452 |
| Type 3 (%) | 3.23 | 14.29 | 0.024 |
| Type 4 (%) | 3.23 | 3.57 | 0.909 |
| Type 5 (%) | 9.68 | 39.29 | <0.001 |

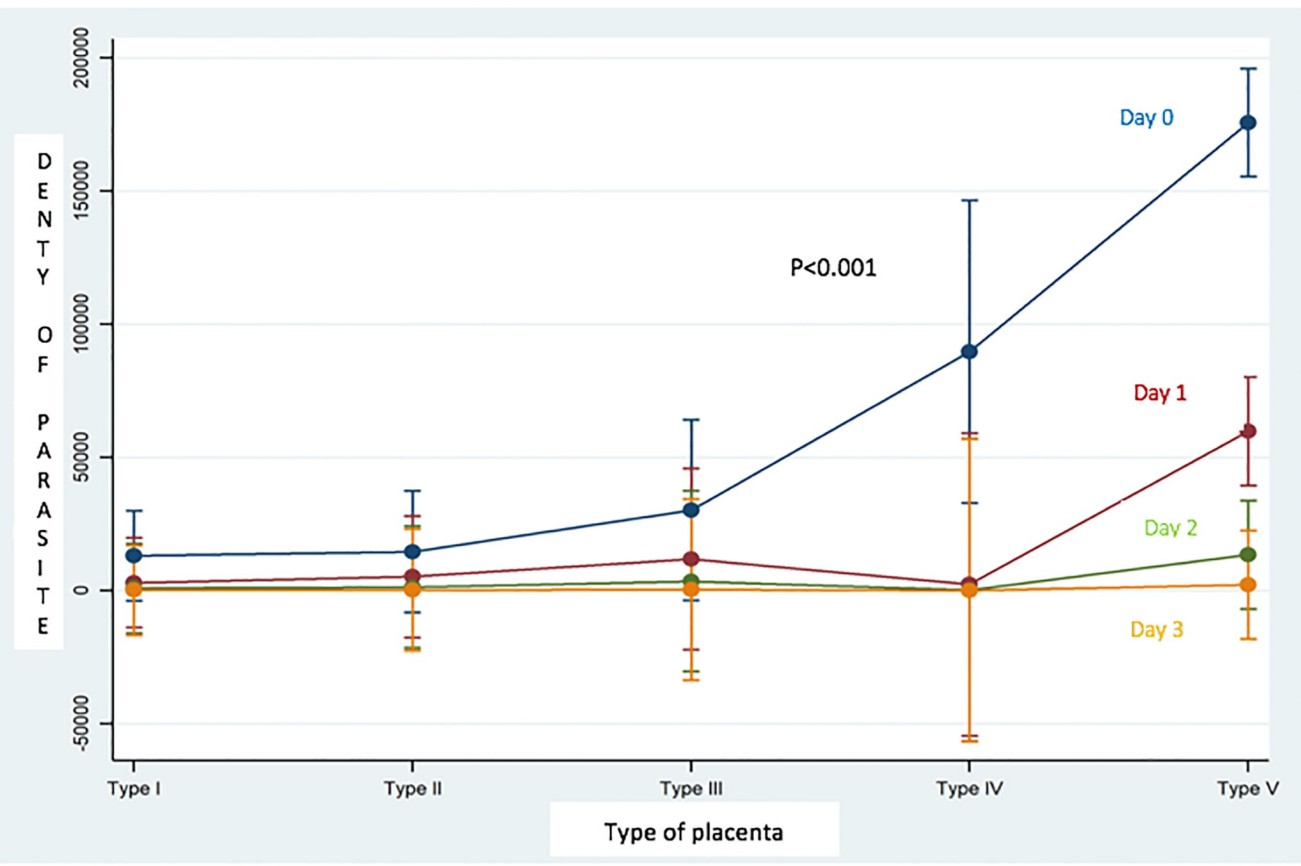

**Fig 3. Correlation between parasite density and type of placental lesion.**

received preventive chemoprophylaxis against malaria. LBW has an impact on the weight of the placenta. Our results showed that low weight placentas with high parasite densities were associated with greater histopathological lesions. The histological lesions encountered in our study (ischemic necrosis and vessel thrombosis) could be directly associated with poor quality exchanges between the mother and the fetus. The disruption of exchanges between the mother and the fetus explains the LBW. Indeed, we observed a positive correlation between the types of placental lesions, parasite density and low birth weight. The etiologies of the histological lesions of the placenta can be the frequency and the severity of the VOCs which increase during the complicated malaria such as disassemble in our study, but also the vasoocclusion of the vessels of the placenta can involve the lesions of ischemic necrosis and thrombosis. vessels. Conversely, the inflammatory phenomenon secondary to vasoocclusion explains inflammatory lesions of the placenta such as chronic villitis. Our study showed that complicated malaria is associated with frequent VOCs and a risk of worsening anemia leading to multiple transfusions during pregnancy.

## Conclusion

Complicated malaria is associated with high maternal and fetal morbidity in sickle cell patients. Malaria chemoprophylaxis can reduce both the parasite density during malaria

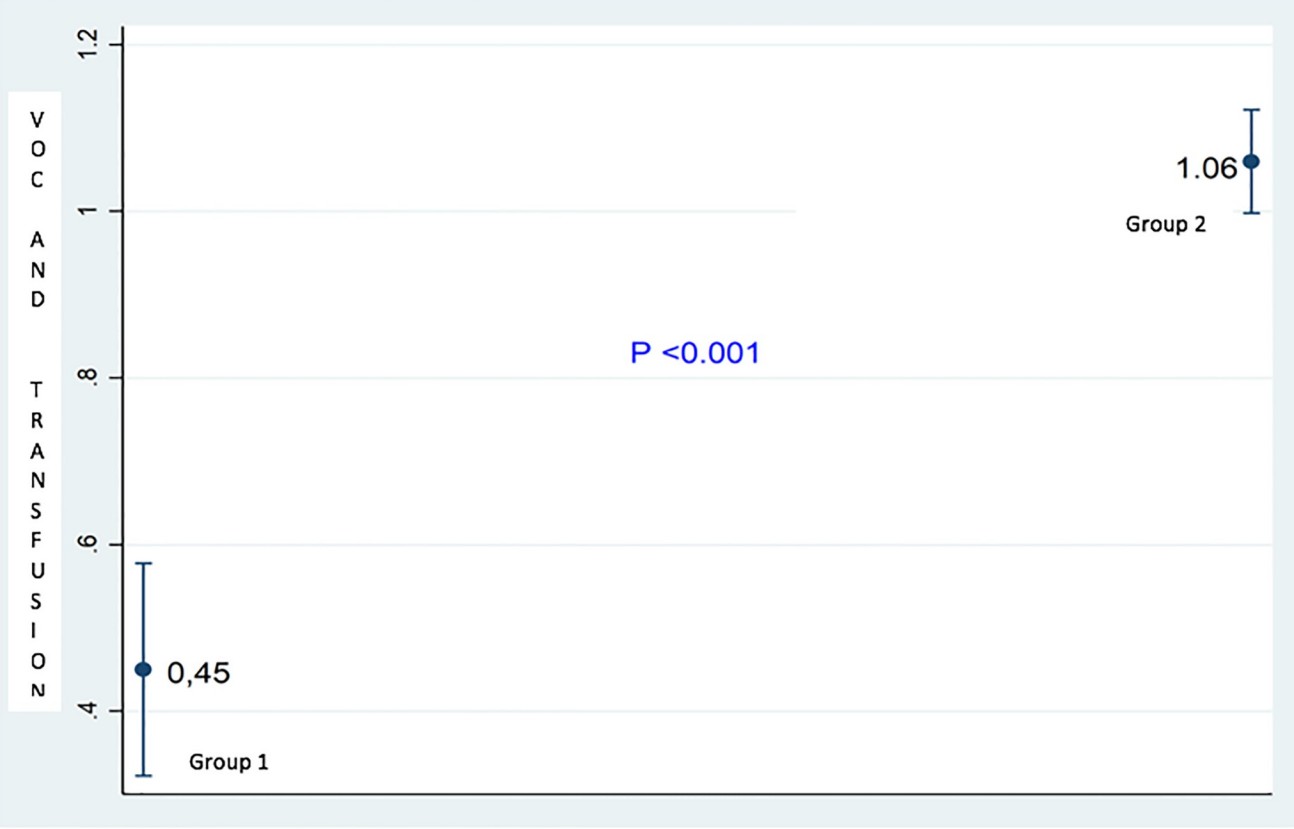

**Fig 4. Frequencies of VOCs and transfusions during the crisis of complicated malaria.**

infection but also reduces the risk of maternal and fetal complications. The risk of low birth weight is associated with damage to the placenta.

## Supporting information

**S1 Data.**
(XLSX)

## Acknowledgments

The authors would like to thank the biologist Jonas KAPAY and his team for the quality of the laboratory analyses. thanks also to the midwife Mamie BAYA and her team for the quality of follow-up of pregnant women during this study. Finally, thanks to all the pregnant women who agreed to participate in this study.

## Author Contributions

**Conceptualization:** Tite Minga Mikobi.

**Data curation:** Tite Minga Mikobi, Pierre Zalagile Akilimali.

**Formal analysis:** Tite Minga Mikobi, Pierre Zalagile Akilimali.

**Investigation:** Tite Minga Mikobi, Nelly Ciombo Kamuanya.

**Methodology:** Tite Minga Mikobi.

**Project administration:** Tite Minga Mikobi.

**Software:** Pierre Zalagile Akilimali.

**Supervision:** Tite Minga Mikobi, Nelly Ciombo Kamuanya, Prosper Tshilobo Lukusa.

**Validation:** Tite Minga Mikobi, Nelly Ciombo Kamuanya, Pierre Zalagile Akilimali, Prosper Tshilobo Lukusa.

**Visualization:** Tite Minga Mikobi, Nelly Ciombo Kamuanya, Pierre Zalagile Akilimali, Prosper Tshilobo Lukusa.

**Writing – original draft:** Tite Minga Mikobi, Prosper Tshilobo Lukusa.

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
