## [Decision Letter · Decision Letter 0]

12 Jul 2022

PONE-D-22-15360SICKLE CELL DISEASE AND PREGNANCY PROFILE OF COMPLICATED MALARIA IN 982 PREGNANCIES IN KINSHASA PLOS ONE

Dear Dr. %LAST_MikobiThank you for submitting your manuscript to PLOS ONE. After careful consideration, we feel that it has merit but does not fully meet PLOS ONE’s publication criteria as it currently stands. Therefore, we invite you to submit a revised version of the manuscript that addresses the points raised during the review process.

We look forward to receiving your revised manuscript.

Kind regards,

Mohamed A Yassin, MD

Academic Editor

PLOS ONE

Journal Requirements:

 No, The funders had no role in study design, data collection and analysis, decision to publish, or preparation of the manuscript.

4. Please ensure that you include a title page within your main document. You should list all authors and all affiliations as per our author instructions and clearly indicate the corresponding author.

6. Please include your tables as part of your main manuscript and remove the individual files. Please note that supplementary tables should remain as separate "supporting information" files.

Reviewers' comments:

Reviewer's Responses to Questions

**Comments to the Author**

1. Is the manuscript technically sound, and do the data support the conclusions?

Reviewer #1: Yes

Reviewer #2: Partly

2. Has the statistical analysis been performed appropriately and rigorously? 

Reviewer #1: N/A

Reviewer #2: I Don't Know

3. Have the authors made all data underlying the findings in their manuscript fully available?

Reviewer #1: Yes

Reviewer #2: Yes

4. Is the manuscript presented in an intelligible fashion and written in standard English?

Reviewer #1: Yes

Reviewer #2: Yes

5. Review Comments to the Author

Reviewer #1: Manuscript Number: PONE-D-22-15360

I have gone through the whole manuscript. In this study authors trying to establish that complicated malaria is associated with high maternal and fetal morbidity in sickle cell patients. Data generated in this study are convincing. The manuscript is well-written and interesting for the scientific and medical reader aspects. I have a few suggestions for the improvement of the manuscript

1)The author is suggested to present the novelty of the conducted research work.

2)The literature study regarding “Sickle cell disease and pregnancy profile of complicated malaria” needs more study backup and more elaborated discussion.

3)Authors needs to discuss more about the potential impact and future direction of this study in terms of therapeutic regimen.

Reviewer #2: The aim of this study is not clear, usually Sickle Cell Diseases are resistance to malaria. why this study had been done? what is their novelty? what was infant health condition? what kind of malaria were studied? what kind of malaria was danger than other?

more discussion is necessary.

6. PLOS authors have the option to publish the peer review history of their article (what does this mean?). If published, this will include your full peer review and any attached files.

Reviewer #1: No

Reviewer #2: No

---

## [Author Response · Author response to Decision Letter 0]

31 Jul 2022

REVIEWER 1

Dear Reviewer

Thank you for the questions of clarification and all your comments in order to improve the quality of our manuscript. Please find the answers to your questions below.

Q1. The author is suggested to present the novelty of the conducted research work.

R1/ In the present study, the results show that complicated malaria in sickle cell pregnant women is associated with high maternal morbidity (low weight gain during pregnancy, increased frequency and severity of acute sickle cell complications: VOC and haemolysis). The frequency as well as the severity of these complications are correlated to the parasite density during complicated malaria. In the fetus and placenta, birth weight, placental weight and observed placental lesions are correlated with parasite density. Finally, chemoprophylaxis can reduce the parasite density during pregnancy.

Q2. The literature study regarding “Sickle cell disease and pregnancy profile of complicated malaria” needs more study backup and more elaborated discussion.

R2/ Dear reviewer, thank you for this contribution, we have enriched the discussion in this revised version

Q3. Authors needs to discuss more about the potential impact and future direction of this study in terms of therapeutic regimen.

R3/ Dear reviewer, thank you for this important observation. Thus, in the revised version of the manuscript we proposed to generalize chemoprophylaxis against malaria in all sickle cell pregnant women.

REVIEWER 2.

Dear Reviewer

Thank you for the questions of clarification and all your comments in order to improve the quality of our manuscript. Please find the answers to your questions below.

Q1. The aim of this study is not clear

R1/ In the present study we have evaluated the morbidity of complicated malaria in sickle cell pregnant women and in the fetus.

Our results show that complicated malaria has a negative impact on maternal health (low pregnancy weight gain, increased frequency and severity of acute sickle cell complications: VOC and haemolysis)

In the fetus, we observed a low birth weight, more significant placental lesions in group 2 (pregnant women who did not receive chemoprophylaxis) than in group 1 (pregnant women who received chemoprophylaxis).

Q2. Usually Sickle Cell Diseases are resistance to malaria. why this study had been done?

R2/ It has been established that only heterozygotes (AS) are protected against severe forms of malaria, whereas homozygotes (SS) are exposed to the severe form of malaria. Malaria is the leading cause of death for children with sickle cell disease under the age of 5.

Q3. what is their novelty?

R3/ the results of our study show that acute maternal sickle cell complications (VOC, haemolysis) increase in frequency and severity during complicated malaria and are correlated with parasite density. Types of placental lesions are also correlated with parasite density

Q4. what was infant health condition?

R4/ the results of our study show very low birth weights and low placenta weights. These low birth weights and placentas are correlated with the types of lesions observed in the placentas.

Q5. what kind of malaria were studied?

R5/ We have studied complicated malaria with a high parasite density

Q6. what kind of malaria was danger than other?

R6/ In sub-Saharan Africa, plasmodium falciparum is the only species associated with complicated malaria. The latter can cause simple malaria, generally benign without great morbidity, or complicated malaria responsible for the mortality reported in sub-Saharan Africa, especially in children under 5 and pregnant women.

Q7. more discussion is necessary.

R7/ Dear reviewer, thank you for this contribution, we have enriched the discussion in this revised version

---

## [Editor Report · Decision Letter 1]

12 Sep 2022

SICKLE CELL DISEASE AND PREGNANCY PROFILE OF COMPLICATED MALARIA IN 982 PREGNANCIES IN KINSHASA

PONE-D-22-15360R1

Dear Dr. Mikobi,

We’re pleased to inform you that your manuscript has been judged scientifically suitable for publication and will be formally accepted for publication once it meets all outstanding technical requirements.

Kind regards,

Mohamed A Yassin, MD

Academic Editor

PLOS ONE
---

## [Editor Report · Acceptance letter]

19 Sep 2022

PONE-D-22-15360R1 

Sickle cell disease and pregnancy profile of complicated malaria in 982 pregnancies in Kinshasa 

Dear Dr. Mikobi:

I'm pleased to inform you that your manuscript has been deemed suitable for publication in PLOS ONE. Congratulations! Your manuscript is now with our production department. 

Kind regards, 

on behalf of

Dr. Mohamed A Yassin 

Academic Editor

PLOS ONE